# DragDiffusion: Harnessing Diffusion Models for Interactive Point-based Image Editing

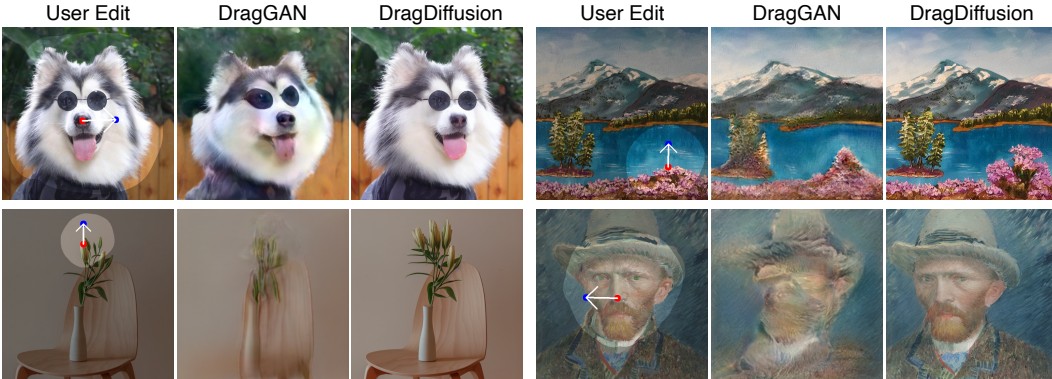

Figure 1: DRAGDIFFUSION **greatly improves the applicability of interactive point-based editing.** Given an input image, the user clicks handle points (red), target points (blue), and draws a mask specifying the editable region (brighter area).

## Abstract

Accurate and controllable image editing is a challenging task that has attracted significant attention recently. Notably, DRAGGAN (Pan et al., 2023) is an interactive point-based image editing framework that achieves impressive editing results with pixel-level precision. However, due to its reliance on generative adversarial networks (GANs), its generality is limited by the capacity of pretrained GAN models. In this work, we extend this editing framework to diffusion models and propose a novel approach DRAGDIFFUSION. By harnessing large-scale pretrained diffusion models, we greatly enhance the applicability of interactive point-based editing on both real and diffusion-generated images. Our approach involves optimizing the diffusion latents to achieve precise spatial control. The supervision signal of this optimization process is from the diffusion model's UNet features, which are known to contain rich semantic and geometric information. Moreover, we introduce two additional techniques, namely LoRA fine-tuning and latent-MasaCtrl, to further preserve the identity of the original image. Lastly, we present a challenging benchmark dataset called DRAGBENCH—the first benchmark to evaluate the performance of interactive point-based image editing methods. Experiments across a wide range of challenging cases (e.g., images with multiple objects, diverse object categories, various styles, etc.) demonstrate the versatility and generality of DRAGDIFFUSION. Code and dataset will be released.

## 1 Introduction

Image editing with generative models (Roich et al., 2022; Endo, 2022; Hertz et al., 2022; Mokady et al., 2023; Kawar et al., 2023; Parmar et al., 2023) has attracted extensive attention recently. One landmark work is DRAGGAN (Pan et al., 2023), which enables interactive point-based image editing, *i.e.*, drag-based editing. Under this framework, the user first clicks several pairs of handle and target points on an image. Then, the model performs semantically coherent editing on the image that moves the contents of the handle points to the corresponding target points. In addition, users can draw a mask to specify which region of the image is editable while the rest remains unchanged.

Despite DRAGGAN's impressive editing results with pixel-level spatial control, the applicability of this method is being limited by the inherent model capacity of generative adversarial networks (GANs) (Goodfellow et al., 2014; Karras et al., 2019; 2020). On the other hand, although large-scale text-to-image diffusion models (Rombach et al., 2022; Saharia et al., 2022) have convincingly demonstrated strong capabilities to synthesize high quality images across various domains, there are not many diffusion-based editing methods that can achieve precise spatial control. This is because most diffusion-based methods (Hertz et al., 2022; Mokady et al., 2023; Kawar et al., 2023; Parmar et al., 2023) conduct editing by controling the text embeddings, which makes their applications primarily focus on editing high-level semantic contents or styles.

To bridge this gap, we propose DRAGDIFFUSION, the first interactive point-based image editing method with diffusion models (Rombach et al., 2022; Saharia et al., 2022; Sohl-Dickstein et al., 2015; Ho et al., 2020). Empowered by large-scale pre-trained diffusion models (Rombach et al., 2022; Saharia et al., 2022), DRAGDIFFUSION achieves accurate spatial control in image editing with significantly better generalizability (see Fig. 1). Our approach is mainly inspired by two important observations from the previous literature: 1) diffusion latents can accurately determine the spatial layout of the generated images (Mao et al., 2023); 2) the UNet output feature maps given the diffusion latent encapsulate rich semantic and geometric information (Tang et al., 2023; Zhang et al., 2023). These two observations motivate us to use the UNet output feature maps as the supervision signal and optimize the diffusion latents to achieve drag-based editing. Furthermore, given the Markov chain nature of diffusion latents, we only need to manipulate the latent at one single diffusion step, which makes our method efficient for practical application.

To be more specific, our approach comprises the following three key steps: Firstly, to preserve the image identity across the editing process, we fine-tune a LoRA (Hu et al., 2022) on the diffusion UNet parameters when editing a real image. Secondly, we optimize the diffusion latent of the image using UNet feature maps to achieve the desired editing. Finally, when denoising the optimized latent to obtain the edited image, we apply a self-attention control mechanism called Latent-MasaCtrl. This mechanism, inspired by MasaCtrl (Cao et al., 2023), is employed to further enhance the consistency of image identity between the edited image and the original image. More details about these three steps are elaborated upon in Sec. 3.

It would be ideal to immediately evaluate our method on well-established benchmark datasets. However, due to a lack of evaluation benchmarks for interactive point-based editing, it is hard to rigorously study and corroborate the efficacy of our proposed approach. Therefore, to facilitate such evaluation, we present DRAGBENCH—the first benchmark dataset for drag-based editing. DRAGBENCH is a diverse collection comprising images spanning various object categories, indoor and outdoor scenes, realistic and aesthetic styles, *etc*. Each image in our dataset is accompanied with a set of "drag" instructions, which consists of one or more pairs of handle and target points as well as a mask specifying the editable region.

Through extensive qualitative and quantitative experiments on a variety of examples, we demonstrate the versatility and generality of our DRAGDIFFUSION approach. In addition, our empirical findings substantiate the crucial role played by LoRA fine-tuning and Latent-MasaCtrl. Furthermore, a comprehensive ablation study is conducted to meticulously explore the influence of key hyper-parameters, including the number of diffusion steps of the optimized latent and the number of LoRA fine-tuning steps.

Our contributions are summarized as follows: 1) we present a novel image editing method DRAGDIFFUSION, the first to achieve interactive point-based editing with diffusion models; 2) we introduce DRAGBENCH, the first benchmark dataset to evaluate interactive point-based image editing methods; 3) Comprehensive qualitative and quantitative evaluation demonstrate the versatility and generality of our DRAGDIFFUSION.

## 2 RELATED WORK

**Generative Image Editing.** Given the initial successes of generative adversarial networks (GANs) in image generation (Goodfellow et al., 2014; Karras et al., 2019; 2020), many previous image editing methods have been based on the GAN paradigm (Endo, 2022; Pan et al., 2023; Abdal et al., 2021; Leimkühler & Drettakis, 2021; Patashnik et al., 2021; Shen et al., 2020; Shen & Zhou, 2021;

Tewari et al., 2020; Härkönen et al., 2020; Zhu et al., 2016; 2023). However, due to the limited model capacity of GANs and the difficulty of inverting the real images into GAN latents (Abdal et al., 2019; Creswell & Bharath, 2018; Lipton & Tripathi, 2017; Roich et al., 2022), the generality of these methods would inevitably be constrained. Recently, due to the impressive generation results from large-scale text-to-image diffusion models (Rombach et al., 2022; Saharia et al., 2022), many diffusion-based image editing methods have been proposed (Hertz et al., 2022; Cao et al., 2023; Mao et al., 2023; Kawar et al., 2023; Parmar et al., 2023; Liew et al., 2022; Mou et al., 2023; Tumanyan et al., 2023; Brooks et al., 2023; Meng et al., 2021; Bar-Tal et al., 2022). Most of these methods aim to edit the images by manipulating the prompts accompanied with the image. However, as many editing attempts are difficult to convey through text, the prompt-based paradigm usually alters the image's high-level semantics or styles, lacking the capability of achieving precise pixel-level spatial control. Epstein et al. (2023) is one of the early efforts in exploring better controllability on diffusion models beyond the prompt-based image editing. In our work, we aim at enabling a even more versatile paradigm than the one studied in Epstein et al. (2023) with diffusion models—interactive point-based image editing.

**Point-based editing.** To enable fine-grained editing, several works have been proposed to perform point-based editing, such as Pan et al. (2023); Endo (2022); Wang et al. (2022). In particular, DRAG-GAN has demonstrated impressive dragging-based manipulation with two simple ingredients: 1) optimization of latent codes to move the handle points towards their target locations and 2) a point tracking mechanism that keep tracks of the handle points. However, its generality is constrained due to the limited capacity of GAN. FreeDrag (Ling et al., 2023) propose to improve DRAGGAN by introducing a point-tracking-free paradigm. In this work, we extend the editing framework of DRAGGAN to diffusion models and showcase its generality over different domains. There is a work (Mou et al., 2023) concurrent to ours that also studies drag-based editing with diffusion models. Differently, they rely on classifier guidance to transforms the editing signal into gradients.

**LoRA in Diffusion Models.** Low Rank Adaptation (*i.e.*, LoRA) (Hu et al., 2022) is a general technique to conduct parameter-efficient fine-tuning on large and deep networks. During LoRA fine-tuning, the original weights of the model are frozen, while trainable rank decomposition matrices are injected into each layer. The core assumption of this strategy is that the model weights will primarily be adapted within a low rank subspace during fine-tuning. While LoRA was initially introduced for adapting language models to downstream tasks, recent efforts have illustrated its effectiveness when applied in conjunction with diffusion models (Ryu, 2022; Gu et al., 2023). In this work, inspired by the promising results of fine-tuning the Stable Diffusion UNet for subject-driven image generation (Ruiz et al., 2023) and prompt-based image editing (Kawar et al., 2023), we also fine-tune a LoRA on the UNet to preserve image identity across the editing process.

## 3 METHODOLOGY

In this section, we formally present the proposed DRAGDIFFUSION approach. To commence, we introduce the preliminaries on diffusion models. Then, we elaborate on the three key stages of our approach as depicted in Fig. 2: 1) fine-tuning a LoRA to better reconstruct the given real image; 2) latent optimization according to the user-provided dragging instructions; 3) denoising the optimized latent guided by our Latent-MasaCtrl mechanism.

### 3.1 PRELIMINARIES ON DIFFUSION MODELS

Denoising diffusion probabilistic models (DDPM) (Sohl-Dickstein et al., 2015; Ho et al., 2020) constitute a family of latent generative models. Concerning a data distribution $q(z)$, DDPM approximates $q(z)$ as the marginal $p_\theta(z_0)$ of the joint distribution between $Z_0$ and a collection of latent random variables $Z_{1:T}$. Specifically,

$$p_\theta(z_0) = \int p_\theta(z_{0:T}) \, \mathrm{d}z_{1:T}, \tag{1}$$

where $p_\theta(z_T)$ is a standard normal distribution and the transition kernels $p_\theta(z_{t-1}|z_t)$ of this Markov chain are all Gaussian conditioned on $z_t$. In our context, $Z_0$ corresponds to image samples given by users, and $Z_t$ corresponds to the latent after $t$ steps of the diffusion process.

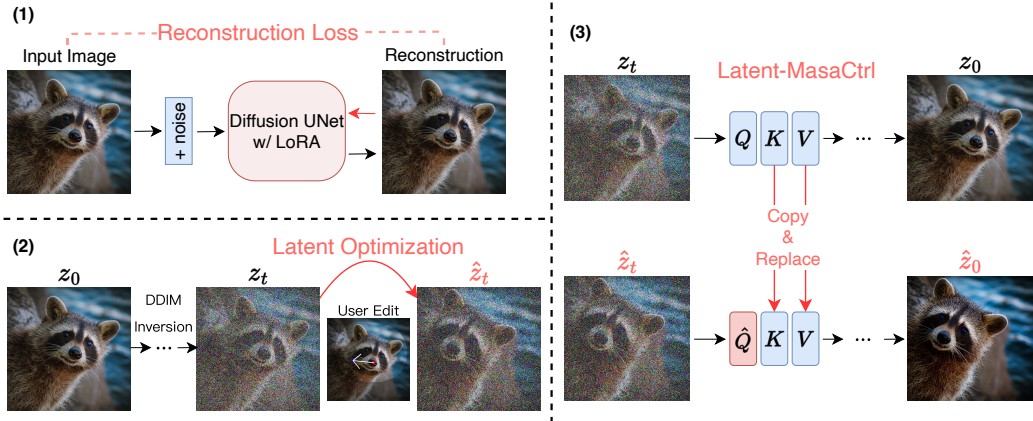

Figure 2: **Overview of** DRAGDIFFUSION. Our approach constitutes three steps: firstly, we fine-tune a LoRA on the UNet of the diffusion model given the input image. Secondly, according to the user's dragging instruction, we optimize the latent obtained from DDIM inversion on the input image. Thirdly, we apply DDIM denoising guided by Latent-MasaCtrl mechanism on $\hat{z}_t$ to obtain the final editing result $\hat{z}_0$. Figure best viewed in color.

Rombach et al. (2022) proposed the latent diffusion model (LDM), which maps data into a lower-dimensional space via a variational auto-encoder (VAE) (Kingma & Welling, 2013) and models the distribution of the latent embeddings instead. Based on the framework of LDM, several powerful pretrained diffusion models have been released publicly, including the Stable Diffusion (SD) model (https://huggingface.co/stabilityai). In SD, the network responsible for modeling $p_\theta(z_{t-1}|z_t)$ is implemented as a UNet (Ronneberger et al., 2015) that comprises multiple self-attention and cross-attention modules (Vaswani et al., 2017). Applications in this paper are implemented based on the public Stable Diffusion model.

### 3.2 LORA FINE-TUNING

Before editing a real image, we first conduct LoRA fine-tuning (Hu et al., 2022) on the diffusion models' UNet (see Fig. 2 (1)). This stage aims to ensure that the diffusion UNet encodes the features of input image more accurately (than in the absence of this procedure), thus facilitating the preservation of the identity of the image throughout the editing process. More formally, the objective function of LoRA is

$$\mathcal{L}_{\text{ft}}(z, \Delta\theta) = \mathbb{E}_{\epsilon, t}\left[\|\epsilon - \epsilon_{\theta+\Delta\theta}(\alpha_t z + \sigma_t \epsilon)\|_2^2\right], \tag{2}$$

where $\theta$ and $\Delta\theta$ represent the UNet and LoRA parameters respectively, $z$ is the real image, $\epsilon \sim \mathcal{N}(\mathbf{0}, \mathbf{I})$ is the randomly sampled noise map, $\epsilon_{\theta+\Delta\theta}(\cdot)$ is the noise map predicted by the LoRA-integrated UNet, and $\alpha_t$ and $\sigma_t$ are parameters of the diffusion noise schedule at diffusion step $t$. The fine-tuning objective is optimized via gradient descent on $\Delta\theta$.

To enable efficient tuning of the parameters, we only incorporate LoRA into the projection matrices of the query, key, and value of attention modules in the diffusion UNet. Moreover, unlike tasks such as subject-driven image generation, which normally fine-tunes the LoRA for around $1,000$ steps, we empirically find that fine-tuning LoRA for only 200 steps is sufficient for our task. This ensures that our LoRA fine-tuning process is extremely efficient.

### 3.3 LATENT OPTIMIZATION

After LoRA fine-tuning, we optimize the diffusion latent according to the user instruction (*i.e.*, the handle and target points, and optionally a mask specifying the editable region) to achieve the desired interactive point-based editing (see Fig. 2 (2)).

Our latent optimization strategy is motivated by the observations that diffusion latents can accurately control the spatial layout of the generated images (Mao et al., 2023). Moreover, previous successes of using UNet feature maps for semantic correspondence (Tang et al., 2023; Zhang et al., 2023) have

demonstrated that the rich semantic and geometric information are encoded in the UNet feature maps. Therefore, we use these feature to supervise the latent optimization process. Moreover, considering the Markov chain structure of diffusion latents, we focus on optimizing the latent of one single diffusion step, thereby attaining both efficiency and efficacy during the editing.

To commence, we first apply DDIM inversion (Song et al., 2021) on the given real image to obtain a diffusion latent at a certain step $t$ (*i.e.*, $z_t$). This diffusion latent serves as the initial value for our latent optimization process. Then, following along the similar spirit of Pan et al. (2023), the latent optimization process consists of two steps to be implemented consecutively. These two steps are motion supervision and point tracking and are executed repeatedly until either all handle points have moved to the targets or the maximum number of iterations has been reached. Next, we describe these two steps in detail.

**Motion Supervision:** We denote the $n$ handle points at the $k$-th motion supervision iteration as $\{h_i^k = (x_i^k, y_i^k) : i = 1, \ldots, n\}$ and their corresponding target points as $\{g_i = (\tilde{x}_i, \tilde{y}_i) : i = 1, \ldots, n\}$. The input image is denoted as $z_0$; the $t$-th step latent (*i.e.*, result of $t$-th step DDIM inversion) is denoted as $z_t$. We denote the UNet output feature maps used for motion supervision as $F(z_t)$, and the feature vector at pixel location $h_i^k$ as $F_{h_i^k}(z_t)$. Also, we denote the square patch centered around $h_i^k$ as $\Omega(h_i^k, r_1) = \{(x, y) : |x - x_i^k| \leq r_1, |y - y_i^k| \leq r_1\}$. Then, the motion supervision loss at the $k$-th iteration is defined as:

$$\mathcal{L}_{\mathrm{ms}}(\hat{z}_t^k) = \sum_{i=1}^{n} \sum_{q \in \Omega(h_i^k, r_1)} \left\| F_{q+d_i}(\hat{z}_t^k) - \mathrm{sg}(F_q(\hat{z}_t^k)) \right\|_1 + \lambda \left\| (\hat{z}_{t-1}^k - \mathrm{sg}(\hat{z}_{t-1}^0)) \odot (\mathbb{1} - M) \right\|_1, \quad (3)$$

where $\hat{z}_t^k$ is the $t$-th step latent after the $k$-th update, $\mathrm{sg}(\cdot)$ is the stop gradient operator (*i.e.*, the gradient will not be back-propagated for the term $\mathrm{sg}(F_q(\hat{z}_t^k))$), $d_i = (g_i - h_i^k)/\|g_i - h_i^k\|_2$ is the normalized vector pointing from $h_i^k$ to $g_i$, $M$ is the binary mask specified by the user, $F_{q+d_i}(\hat{z}_t^k)$ is obtained via bilinear interpolation as the elements of $q + d_i$ may not be integers. In each iteration, $\hat{z}_t^k$ is updated by taking one gradient descent step to minimize $\mathcal{L}_{\mathrm{ms}}$:

$$\hat{z}_t^{k+1} = \hat{z}_t^k - \eta \cdot \frac{\partial \mathcal{L}_{\mathrm{ms}}(\hat{z}_t^k)}{\partial \hat{z}_t^k}, \quad (4)$$

where $\eta$ is the learning rate for latent optimization.

Note that for the second term in Eqn. (3), which encourages the unmasked area to remain unchanged, we are working with the diffusion latent instead of the UNet features. Specifically, given $\hat{z}_t^k$, we first apply one step of DDIM denoising to obtain $\hat{z}_{t-1}^k$, then we regularize the unmasked region of $\hat{z}_{t-1}^k$ to be the same as $\hat{z}_{t-1}^0$ (*i.e.*, $z_{t-1}$).

**Point Tracking:** Since the motion supervision updates $\hat{z}_t^k$, the positions of the handle points may also change. Therefore, we need to perform point tracking to update the handle points after each motion supervision step. To achieve this goal, we use UNet feature maps $F(\hat{z}_t^{k+1})$ and $F(z_t)$ to track the new handle points. Specifically, we update each of the handle points $h_i^k$ with a nearest neighbor search within the square patch $\Omega(h_i^k, r_2) = \{(x, y) : |x - x_i^k| \leq r_2, |y - y_i^k| \leq r_2\}$ as follows:

$$h_i^{k+1} = \arg\min_{q \in \Omega(h_i^k, r_2)} \left\| F_q(\hat{z}_t^{k+1}) - F_{h_i^0}(z_t) \right\|_1. \quad (5)$$

### 3.4 LATENT-MASACTRL

After we have completed the optimization of the diffusion latents, we then denoise the optimized latents to obtain the final editing results. However, we find that naïvely applying DDIM denoising on the optimized latents still occasionally leads to undesired identity shift and degradation in quality compared to the original image. We posit that this issue arises due to the absence of adequate guidance from the original image during the denoising process.

To mitigate this problem, we draw inspiration from MasaCtrl (Cao et al., 2023), a prompt-based image editing method that can well preserve the identity of the original image. This method elegantly leverages the property of self-attention modules to steer the editing process, thereby boosting coherence between the original image and the editing results. However, since our approach focuses on

optimizing the latent instead of changing prompt to perform editing, MasaCtrl is not readily applicable in our case. Hence, we derive a new variant of the original MasaCtrl, named Latent-MasaCtrl, which can be seamlessly integrated into our DRAGDIFFUSION pipeline.

In particular, as illustrated in Fig. 2 (3), given the denoising process of both the original latent $z_t$ and the optimized latent $\hat{z}_t$, we use the process of $z_t$ to guide the process of $\hat{z}_t$. More specifically, during the forward propagation of the UNet's self-attention modules in the denoising process, we replace the key and value vectors generated from $\hat{z}_t$ with the ones generated from $z_t$. With this simple replacement technique, the query vectors generated from $\hat{z}_t$ will be directed to query the correlated contents and texture of $z_t$. This result in the denoising results of $\hat{z}_t$ (*i.e.*, $\hat{z}_0$) being more coherent with the denoising results of $z_t$ (*i.e.*, $z_0$). In this way, Latent-MasaCtrl substantially improves the consistency between the original image and our editing results.

## 4 EXPERIMENTS

### 4.1 IMPLEMENTATION DETAILS

In all our experiments, unless stated otherwise, we adopt the Stable Diffusion 1.5 (Rombach et al., 2022) as our diffusion model. As elaborated upon in Sec. 3.2, we inject LoRA into the projection matrices of query, key and value in all of the attention modules. We set the rank of the LoRA to 16. We fine-tune the LoRA using the AdamW (Kingma & Ba, 2015) optimizer with a learning rate of $2 \times 10^{-4}$ for 200 steps before perform drag-based editing.

During the latent optimization stage, we schedule 50 steps for DDIM and optimize the diffusion latent at the 35-th step unless specified otherwise. When editing real images, we *do not* apply classifier-free guidance (CFG) (Ho & Salimans, 2021) in both DDIM inversion and DDIM denoising process. This is because CFG tends to amplify numerical errors, which is not ideal in performing the DDIM inversion (Mokady et al., 2023). We use the Adam optimizer with a learning rate of $0.01$ to optimize the latent. The maximum optimization step is set to be $80$. The hyperparameter $r_1$ in Eqn. 3 and $r_2$ in Eqn. 5 are tuned to be 1 and 3, respectively. $\lambda$ in Eqn. 3 is set to $0.1$ by default, but the user may increase $\lambda$ if the unmasked region has changed to be more than what was desired.

Finally, following Cao et al. (2023), we apply our Latent-MasaCtrl in the upsampling blocks of the diffusion UNet. In addition, we apply Latent-MasaCtrl guidance at all denoising steps when generating the editing results.

### 4.2 DRAGBENCH AND EVALUATION METRICS

Since interactive point-based image editing is a recently introduced paradigm, there is an absence of dedicated evaluation benchmarks for this task, making it challenging to comprehensively study the effectiveness of our proposed approach. To address the need for systematic evaluation, we introduce DRAGBENCH, the first benchmark dataset tailored for drag-based editing. DRAGBENCH is a diverse compilation encompassing various types of images. Details about our dataset is given in Appendix A. Each image within our dataset is accompanied by a set of dragging instructions, comprising one or more pairs of handle and target points, along with a mask indicating the editable region. We hope future research on this task can benefit from DRAGBENCH.

In this work, we utilize the following two metrics for quantitative evaluation: *Image Fidelity* (IF) (Kawar et al., 2023) and *Mean Distance* (MD) (Pan et al., 2023). IF, the first metric, quantifies the similarity between the original and edited images. It is calculated by subtracting the mean LPIPS (Zhang et al., 2018) over all pairs of original and edited images from $1$. The second metric MD assesses how well the approach moves the semantic contents to the target points. To compute the MD, we first employ DIFT (Tang et al., 2023) to identify points in the edited images corresponding to the handle points in the original image. These identified points are considered to be the final handle points post-editing. MD is subsequently computed as the mean Euclidean distance between positions of all target points and their corresponding final handle points. MD is averaged over all pairs of handle and target points in the dataset. An optimal "drag" approach ideally achieves both a low MD (indicating effective "dragging") and a high IF (reflecting robust identity preservation).

|  | User Edit | Drag Results | User Edit | Drag Results | User Edit | Drag Results |

Figure 3: Editing results on real and diffusion-generated images with Stable Diffusion 1.5 (SD-1.5) and its fine-tuned variants. **(a)** Editing real images with SD-1.5. **(b)** Editing generated images from SD-1.5. **(c)** Editing generated images from Counterfeit-V2.5. **(d)** Editing generated images from Majicmix Realistic. **(e)** Editing generated images from Realistic Vision. **(f)** Editing generated images from Interior Design Supermix. **(g)** Editing generated images from DVarch.

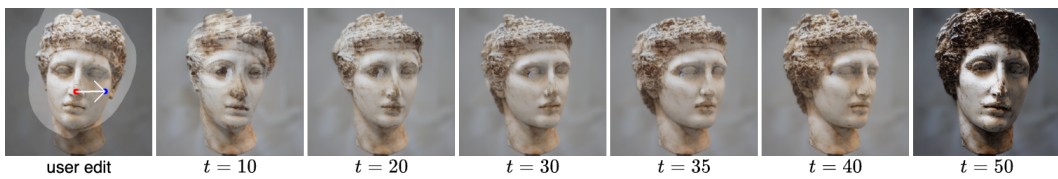

user edit      $t = 10$      $t = 20$      $t = 30$      $t = 35$      $t = 40$      $t = 50$

Figure 4: Ablating the number of inversion step $t$. Effective results are obtained when $t \in [30, 40]$.

### 4.3 QUALITATIVE EVALUATION

In this section, we perform qualitative evaluation on DRAGDIFFUSION, highlighting its capabilities through a visual exploration of outcomes obtained from both real and diffusion-generated images. To present a comprehensive assessment, we employ Stable Diffusion 1.5 (SD-1.5) (Rombach et al., 2022) for all results pertaining to real images. On the other hand, for generated images, we extend our evaluation across a spectrum of fine-tuned variants of SD-1.5, including Counterfeit-V2.5, Majicmix Realistic, Realistic Vision, Interior Design Supermix, and DVarch.

Our visualized outcomes are shown in Fig. 3. As illustrated in the figure, our DRAGDIFFUSION achieves drag-based editing with high quality, both on real-world and diffusion-generated images. Notably, our approach is versatile in that it can deal with images from various domains as well as drag instructions of different magnitudes (*e.g.*, small magnitude edits such as the left-most image in Fig. 3 (e) and large magnitude edits such as the left-most image in Fig. 3 (d)). These results not only underscore DRAGDIFFUSION's ability to produce effective drag-based edits, but also highlight its ability to maintain *consistency* of the edited images with the original ones. More qualitative results can be found in Appendix G.

### 4.4 QUANTITATIVE ANALYSIS

In this section, we conduct a rigorous quantitative evaluation to assess the performance of our approach. We begin by comparing DRAGDIFFUSION with the baseline method DRAGGAN. As each StyleGAN (Karras et al., 2020) model utilized in Pan et al. (2023) is specifically designed for a particular image class, we employ an ensemble strategy to evaluate DRAGGAN. This strategy involves assigning a text description to characterize the images generated by each StyleGAN model. Before editing each image, we compute the CLIP similarity (Radford et al., 2021) between the image and each of the text descriptions associated with the GAN models. The GAN model that yields the highest CLIP similarity is selected for the editing task.

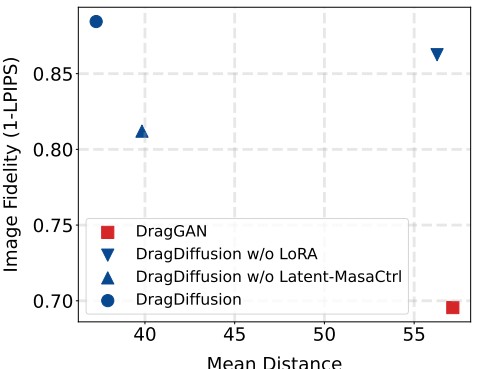

Figure 5: Quantitative analysis on DRAGGAN, DRAGDIFFUSION and DRAGDIFFUSION's variants without certain crucial components. Image Fidelity (↑) and Mean Distance (↓) are reported. Results are produced on DRAGBENCH. The approach with better results should locate at the upper-left corner of the coordinate plane.

Furthermore, to assess the effectiveness of each component of our approach, we evaluate DRAGDIFFUSION in the following two configurations: one without Latent-MasaCtrl and the other without LoRA fine-tuning. We perform our empirical studies on the DRAGBENCH dataset, and Image Fidelity (IF) and Mean Distance (MD) of each configuration mentioned above are reported in Fig. 5. All results are averaged over the DRAGBENCH dataset. In this figure, the $x$-axis represents MD and the $y$-axis represents IF, which indicates the method with better results should locate at the upper-left corner of the coordinate plane. The results presented in this figure clearly demonstrate that our DRAGDIFFUSION significantly outperforms the DRAGGAN baseline in terms of both IF and MD. Additionally, these empirical findings corroborate the crucial role played by LoRA fine-tuning and Latent-MasaCtrl in enhancing both identity preservation and accuracy of the drag editing. Visualization on the effects of LoRA fine-tuning and Latent-MasaCtrl are given in Appendix C.

### 4.5 ABLATION STUDY ON THE NUMBER OF INVERSION STEP

Next, we conduct an ablation study to elucidate the impact of varying $t$ (*i.e.*, the number of inversion steps) during the latent optimization stage of DRAGDIFFUSION. To explore this, we set $t$ to be $t = 10, 20, 30, 40, 50$ steps and run our approach on DRAGBENCH to obtain the editing results ($t = 50$ corresponds to the pure noisy latent). We evaluate Image Fidelity (IF) and Mean Distance (MD) for each $t$ value in Fig. 6. All metrics are averaged over the DRAGBENCH dataset.

In terms of the IF, we observe a monotonic decrease as $t$ increases. This trend can be attributed to the enhanced flexibility of the diffusion latent as more steps are inverted.

However, MD exhibits a more complicate behavior. It initially decreases and then starts to increase with higher $t$ values. This behavior highlights the presence of a critical range of $t$ values for effective editing ($t \in [30, 40]$ in our figure). When $t$ is too small, the diffusion latent lacks the necessary flexibility for substantial changes, posing challenges in performing reasonable edits. Conversely, overly large $t$ values result in a diffusion latent that is unstable for editing, leading to difficulties in preserving the original image's identity. Motivated by our empirical results, we chose $t = 35$ as our default setting, as it achieves the lowest MD while maintaining a decent IF.

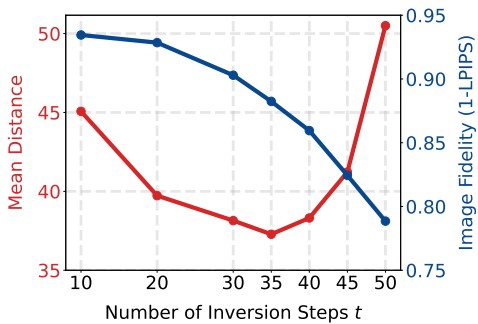

Figure 6: Ablation study on the impact of the inversion step $t$ of the optimized latent. Mean Distance ($\downarrow$) and Image Fidelity ($\uparrow$) are reported. Results are produced on DRAGBENCH.

To complement our quantitative analysis, we provide qualitative visualizations in Fig. 4, which corroborate with the findings obtained through numerical evaluation.

### 4.6 ABLATION STUDY ON THE NUMBER OF LoRA FINE-TUNING STEPS

Finally, we perform an ablation study to examine the influence of varying LoRA fine-tuning steps on the performance of DRAGDIFFUSION when editing real images. Specifically, we acquire six different checkpoints of LoRA fine-tuning, each trained for a different number of steps, namely, 0, 100, 200, 300, 400, and 500 steps, respectively (0 being no LoRA fine-tuning). Subsequently, utilizing each of these LoRA checkpoints, we apply our approach on DRAGBENCH. The outcomes are assessed using IF and MD, and the results are presented in Fig. 7. All results are averaged over the DRAGBENCH dataset. As the number of LoRA fine-tuning steps increases, IF exhibits an increasing trend. This behavior is attributed to the LoRA's enhanced ability to reconstruct the original image with prolonged fine-tuning, resulting in increased consistency with the original image.

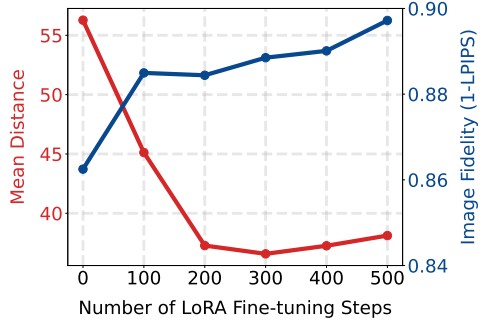

Figure 7: Ablation study on the impact of the number of LoRA fine-tuning steps. Mean Distance ($\downarrow$) and Image Fidelity ($\uparrow$) are reported. Results are produced on DRAGBENCH.

However, the behavior of MD follows a more complicated trajectory. Initially, it experiences a decline, reflecting the limitations of insufficient LoRA fine-tuning, which can lead to pronounced identity shifts and render the "drag" editing ineffective. On the other hand, as the fine-tuning progresses, MD subsequently begins to increase slightly. This phenomenon arises due to an excessive degree of overfitting of the LoRA on the original image when fine-tuned for an extended duration. As a consequence, the edited outcomes tend to adhere closely to the original image, posing challenges to the editing process. Considering the experimental results, we fine-tune LoRA for 200 steps by default to balance between effectiveness and efficiency.

## 5 CONCLUSION AND FUTURE WORKS

In this work, we extend interactive point-based editing to large-scale pretrained diffusion models through the introduction of a novel method named DRAGDIFFUSION. Furthermore, we introduce the DRAGBENCH dataset, which aims to facilitate the evaluation of the interactive point-based editing methods. Comprehensive qualitative and quantitative results showcases the remarkable versatility and generality of our proposed method. Limitations of our approach are further discussed in Appendix F, and we leave making the drag-based editing more robust and reliable on diffusion models as our future work.

ETHICAL STATEMENT

The potential concern of our DRAGDIFFUSION is that one could use our approach to edit real images, which could harm the credibility of real images as evidence, for example, in legal cases.

REPRODUCIBILITY STATEMENT

Implementation details of our DRAGDIFFUSION is provided in Sec. 4.1. Details on our DRAG-BENCH and evaluation metrics are provided in Sec. 4.2 and Appendix A. In addition, all our experiments are performed with publicly available models, and links to the fine-tuned variants of Stable Diffusion being used are given in Appendix B. Most importantly, we have included the code of DRAGDIFFUSION as Supplementary Materials. We do not include our DRAGBENCH for now as the naming patterns of the image files in the dataset might reveal the author identity. However, this dataset will be publicly available in the future.

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

## A    DETAILS ABOUT DRAGBENCH DATASET

Images in our DRAGBENCH are classified into the following 10 categories: animals, art works, buildings (city view), buildings (countryside view), human (face), human (upper body), human (full body), interior design, landscape, other objects. All human-related images are selected from Midjourney generation results to avoid potential legal concerns. All the other images are real images downloaded from unsplash (https://unsplash.com/), pexels (https://www.pexels.com/zh-cn/), and pixabay (https://pixabay.com/).

## B    LINKS TO THE STABLE DIFFUSION'S FINETUNED VARIANTS USED BY US

Here, we provide links to the fine-tuned variants of Stable Diffusion used by us:

Counterfeit-V2.5 (https://huggingface.co/gsdf/Counterfeit-V2.5),

Majixmix Realistic (https://huggingface.co/emilianJR/majicMIX_realistic),

Realistic Vision (https://huggingface.co/SG161222/Realistic_Vision_V2.0),

Interior Design Supermix (https://huggingface.co/stablediffusionapi/interiordesignsuperm),

DVarch (https://huggingface.co/stablediffusionapi/dvarch).

## C    VISUAL ILLUSTRATION ON THE EFFECTS OF LoRA FINE-TUNING AND LATENT-MASACTRL

In the main text, we have quantitatively show the effectiveness of LoRA fine-tuning and Latent MasaCtrl. Here, we give some qualitative examples to corroborate our quantitative results. Visualized outcomes are shown in Fig. 8. From the results, we can see that LoRA fine-tuning primarily help us to correctly encode the information provided in the original image, while Latent-MasaCtrl focus on ensuring the style and texture coherent with the original image.

## D    MORE DETAILS ON EDITING DIFFUSION-GENERATED IMAGES

Here we introduce more details about editing diffusion-generated images. Firstly, different from editing real images, we *do not* need to conduct LoRA fine-tuning before latent optimization. This is because the purpose of LoRA fine-tuning is to help better encode the features of the original image into the diffusion UNet. However, for diffusion-generated images, the image features are already well-encoded as the diffusion model itself can generate these images. In addition, during the latent

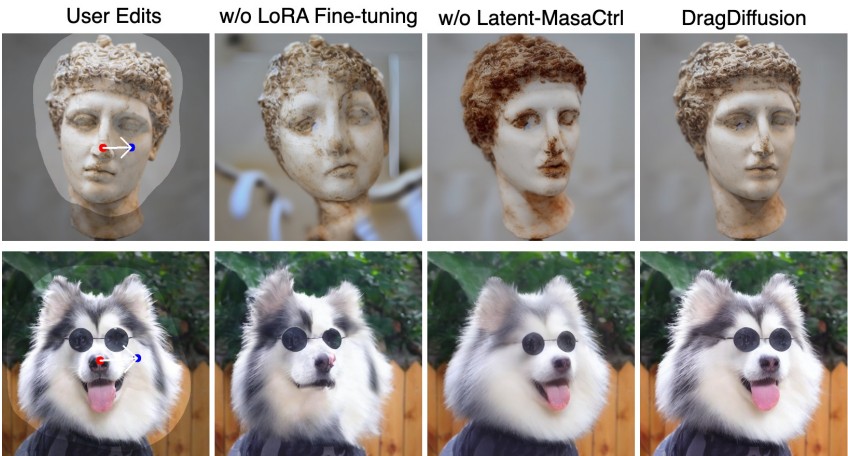

Figure 8: Qualitative validation on LoRA fine-tuning and Latent-MasaCtrl.

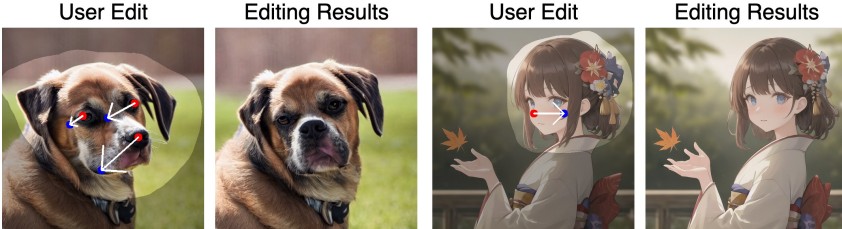

Figure 9: Limitation of DRAGDIFFUSION. Occasionally, some of the handle points cannot precisely reach the desired target.

optimization stage, we do not have to perform DDIM inversion as the diffusion latents are readily available from the generation process of the diffusion models.

Another details we need to attend to is the presence of classifier-free guidance (CFG) when editing generated images. As described in the main text, when editing real images, we turn off the CFG as it pose challenges to DDIM inversion. However, when editing generated images, we inevitably have to deal with CFG, as it is one of the key component in diffusion-based image generation. CFG introduces another forward propagation pass of the UNet during the denoising process with a negative text embedding from null prompt or negative prompt. This makes a difference during our latent optimization stage, as now we have two UNet feature maps (one from the forward propagation with positive text embedding and the other one from the negative text embedding) instead of only one. To deal with this, we concatenate these two feature maps along the channel dimension and then use the combined feature maps to supervise latent optimization. This simple strategy have been proven to be effective as shown in our empirical results.

## E  EXECUTION TIME

Given a real image with the resolution of $512 \times 512$, the execution time of different stages in DRAGDIFFUSION on a A100 GPU is as follows: LoRA fine-tuning is around 45 seconds, latent optimization is around 10 to 30 seconds depending on the magnitude of the drag-instruction, the final Latent-MasaCtrl guided denoising is negligible comparing to previous steps (about 1 to 2 seconds)

## F  LIMITATIONS

As shown in Fig. 9, the limitation of our DRAGDIFFUSION is that, occasionally, some of the handle points cannot precisely reach the desired target. This is potentially due to inaccurate point-tracking or difficulties in latent optimization when multiple pairs of handle and target points are given. We

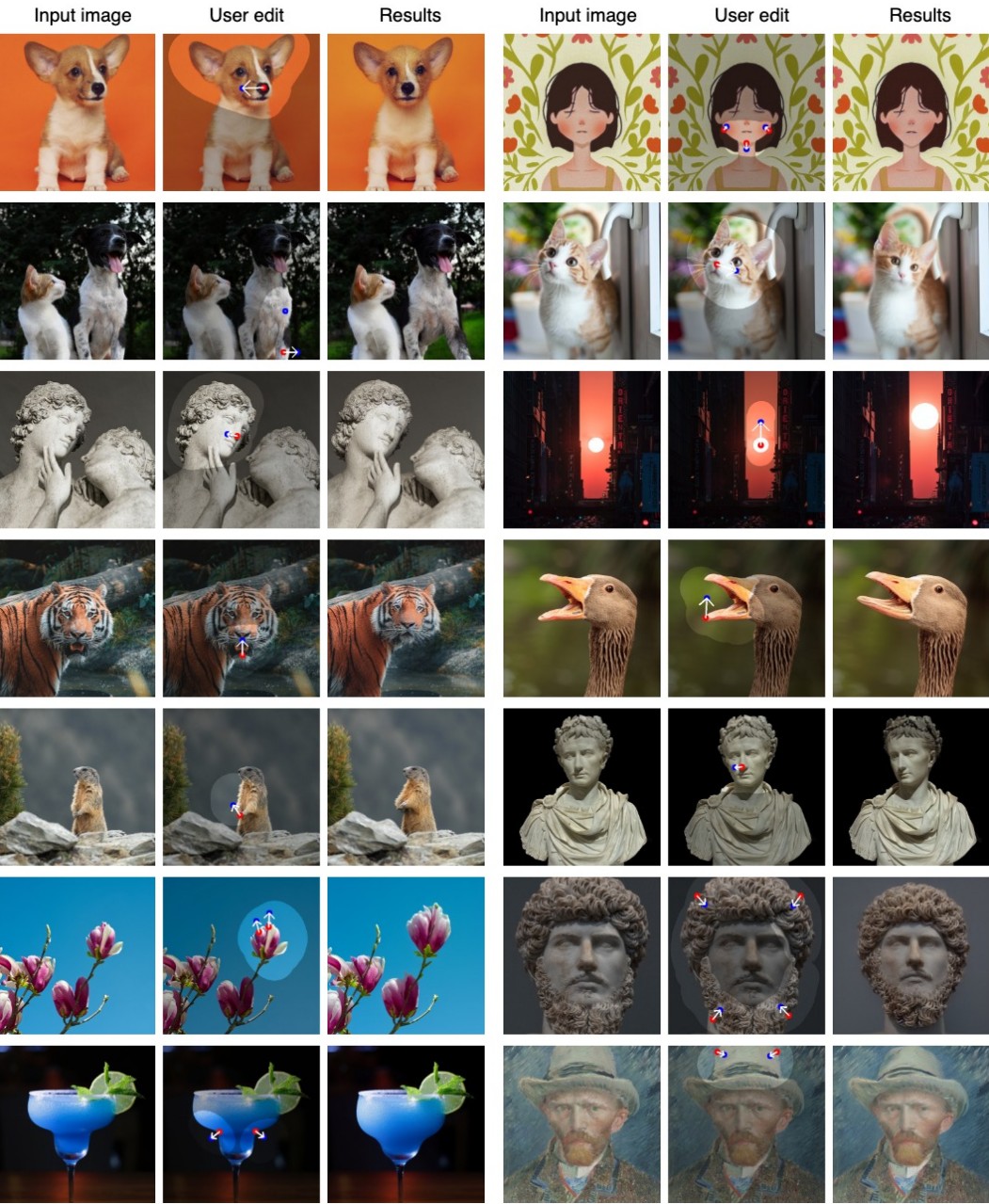

Figure 10: More editing results from DRAGDIFFUSION on real images.

leave making the drag-based editing on diffusion models more robust and reliable as our future work.

## G  MORE QUALITATIVE RESULTS

Here we provide more qualitative results of DRAGDIFFUSION. To start with, provide more editing results on real images in Fig. 10. All real images are edited with Stable Diffusion 1.5. Then, we provide more results on generated images beyond the $512 \times 512$ resolution in the main text. These results are shown in Fig. 11, which further demonstrate the versatility of DRAGDIFFUSION.

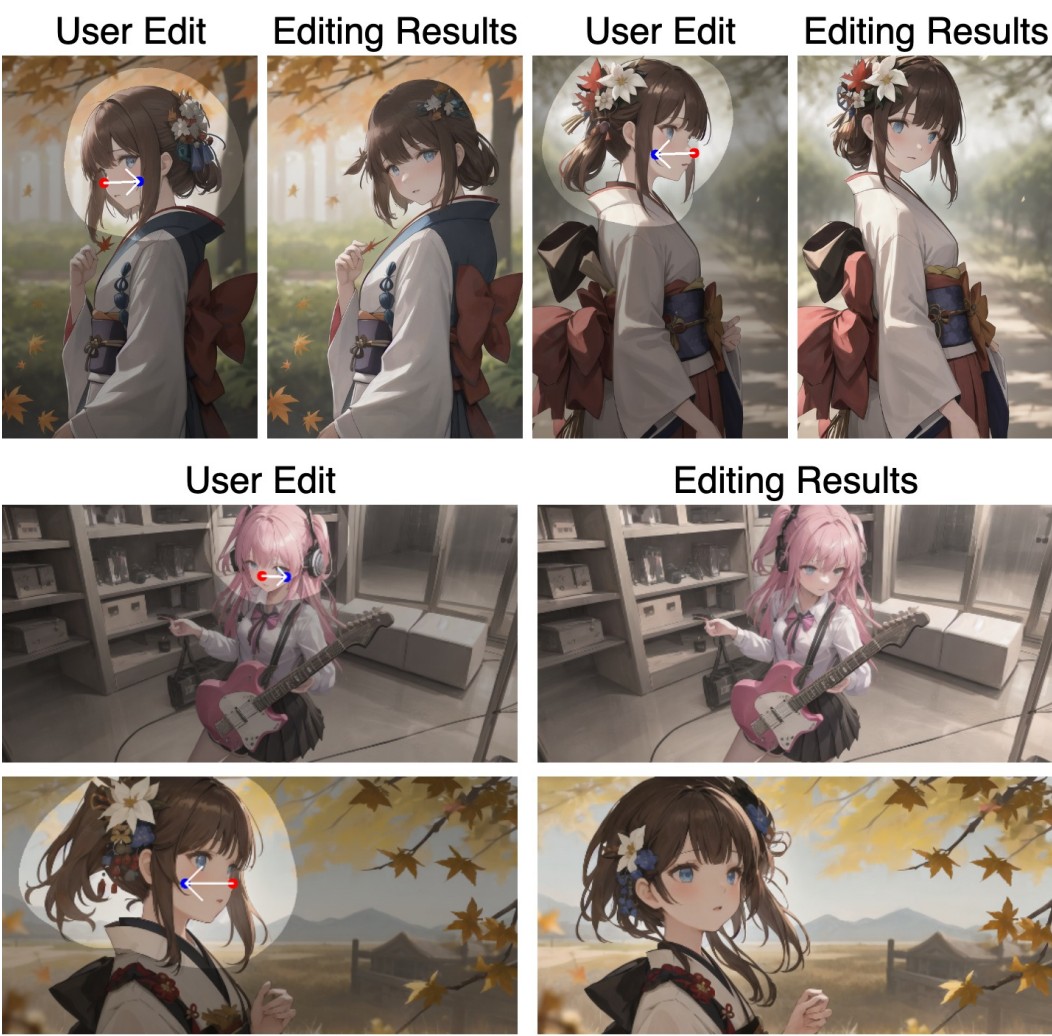

Figure 11: Editing results from DRAGDIFFUSION beyond $512 \times 512$ resolution. Results are produced by perform drag-based edits on images generated by Counterfeit-V2.5. The resolution of images in the first row are $768 \times 512$, while the images in the second row are $512 \times 1024$.

