# OpenReview forum: "DragDiffusion: Harnessing Diffusion Models for Interactive Point-based Image Editing"
_ICLR.cc/2024/Conference — ICLR 2024 Conference Withdrawn Submission_

### Official Review · Reviewer_xNxo · 2023-10-22

**Soundness:** 3 good
**Presentation:** 3 good
**Contribution:** 3 good
**Rating:** 6
**Confidence:** 4

**Summary:**

This paper introduces a point-based image editing technique to pre-trained text-to-image diffusion models. The proposed method consists of a three-stage pipeline: LoRA finetune, point-track based Latent Optimization, and Latent-MasaCtrl denoise process. With the help of an image preserving mask, the proposed method enables reasonable point-based image editing for StableDiffusion models. The authors also introduce a benchmark dataset for drag evaluation.

**Strengths:**

- Point-based image editing is an efficient interactive way for practical image editing applications. This work is one of the first to introduce such editing to diffusion models.

- The proposed method borrows recent advanced techniques to build a reasonable three-stage pipeline.

- Sufficient quantitative results and ablations are constructed to evaluate the proposed components.

- The introduced benchmark dataset can benefit future works.

**Weaknesses:**

- The pipeline design is straightforward, and the design choice analysis would be more useful for future work. For example, there are also other techniques for identity preservation such as id-encoder, null-text inversion, etc.

- The visual comparisons with other baselines (e.g., DragGAN, FreeDrag) are not sufficient. Comparisons on different categories of DragBench should be reported in the appendix. Besides, the comparison with DragGAN in the teaser is not fair since the preserving region is not applied to DragGAN.

- The drag ability introduced in this work seems still very limited: most cases are about local deform or small 3D rotation. Such limitations and failure cases should be discussed in the paper to show the boundary of drag ability.

- The time cost is still very expensive. It is claimed that for a 512x512 image, it may take more than 1 minute on an A100 GPU.

- It would be good to add benchmark data examples in the appendix.

**Questions:**

More analysis of design choices, visual comparisons and limitations are suggested.

---

### Official Review · Reviewer_U8vP · 2023-10-29

**Soundness:** 3 good
**Presentation:** 3 good
**Contribution:** 3 good
**Rating:** 6
**Confidence:** 4

**Summary:**

This work introduces a new point-based image editing technique with high capability and effectiveness, DragDiffusion. The method can be applied to edit a variety of images for point-based editing, including real and synthesized images, as well as challenging cases such as images with multiple objects, surpassing the previous GAN-based editing method DragGAN both quantitatively and qualitatively. A new benchmark for point-based editing evaluation, DragBench, is introduced. It is a beneficial contribution to future research on image editing.

**Strengths:**

- DragDiffusion is of high capability and can effectively accomplish point-based image editing tasks. In some cases, it can produce obviously better and more reasonable results than the DragGAN approach.
- The method is well-designed. Starting from Diffusion UNet LoRA fine-tuning, the method further leverages 1) the latent optimization technique to align latents with user editing, and 2) the latent-MasaCtrl strategy to ensure enough identity between the edited image and the input.
- DragBench is introduced for evaluating point-based image editing.
- Extensive experiments with abundant visual evaluations for illustrating the effectiveness of DragDiffusion. Ablation studies are conducted on the number of inversion steps and the number of LORA fine-tuning steps for revealing the influence of key parameters on the method.

**Weaknesses:**

- The significance of the technical contribution is a little bit limited. Though with insights into the relationship between diffusion latent and the image content, the method relies on existing techniques to acquire good performance, including the latent optimization (from DragGAN) and the latent-MasaCtrl design based on the existing MasaCtrl strategy. Therefore, despite the effectiveness of the method and the smart usage of the insights on diffusion latents, it is hard to tell what new messages or general principles which are potentially useful for other works it can convey.
- The technique is proposed for point-based image editing. It would be better if the method (or with some small modification) could solve other image editing problems. The claim on its strong power, generality, and versatility would be more solid if it could be broadly applied to other editing problems.
- Showing more qualitative comparisons between DragDiffusion and previous or concurrent approaches such as DragGAN would be beneficial to evaluating its superiority in point-based image editing.
- Ablation studies are not particularly sufficient.

DragDiffusion is effective and superior to previous works. However, the main concerns lie in the significance of its technical contribution and the insufficient qualitative comparisons. Thus I vote the acceptance but limited to score 6.

**Questions:**

More visual comparisons?

Other ablation studies?

---

### Official Review · Reviewer_DRgP · 2023-11-01

**Soundness:** 4 excellent
**Presentation:** 4 excellent
**Contribution:** 3 good
**Rating:** 3
**Confidence:** 5

**Summary:**

This paper proposes to enable point-based editing from pretrained Diffusion models like Stable Diffusion. It is inspired from DragGAN. The proposed method extends the point-tracking objectives from DragGAN while performing the sampling from the Diffusion Model. It also employs tricks like LoRA and MasaCtrl to retain the identity of the input image. The results are impressive. Additionally they also create a new benchmark dataset for this task, called DragBench.

**Strengths:**

- Drag based editing in diffusion models with promising results on diverse set of images.
- A dataset to benchmark the drag based editing methods.

**Weaknesses:**

- The approach builds on the existing well established work DragGAN on text-to-image diffusion models. While the results are interesting the method just replicates the same loss function with minor changes (LoRA and MasaCtrl) which are already known in the literature. Hence I believe the technical contribution is limited for the community.
- Discussion on UNet features is not provided. It would be interesting to see an ablation over the intermediate UNet features used for the optimization and how does it affect the edit in the final image.
- Missing experiment - Ablation of using more than one time-step for the drag based optimization. Does it improves the performance of the editing with a better control?
- Details for the DragBench are missing from the main paper and the supplementary. How many images are there in the dataset? How are the labels gathered for the point based control?

**Questions:**

Please refer to the weakness section for questions related to each point.